# The Effect of Saponite Clay on Ruminal Fermentation Parameters during In Vitro Studies

**DOI:** 10.3390/ani14050738

**Published:** 2024-02-27

**Authors:** Alina Pikhtirova, Ewa Pecka-Kiełb, Bożena Króliczewska, Andrzej Zachwieja, Jarosław Króliczewski, Robert Kupczyński

**Affiliations:** 1Department of Public Health, SE Medical Institute, Sumy State University, Rymskogo-Korsakova 2, 40007 Sumy, Ukraine; alinca.sumy@gmail.com; 2Department of Animal Physiology and Biostructure, Wroclaw University of Environmental and Life Sciences, Norwida Str. 31, 50-375 Wroclaw, Poland; bozena.kroliczewska@upwr.edu.pl; 3Animal Breeding, Wroclaw University of Environmental and Life Sciences, ul. Chelmonskiego 38C, 51-631 Wroclaw, Poland; andrzej.zachwieja@upwr.edu.pl; 4Department of Experimental Biology, Wroclaw University of Environmental and Life Sciences, Norwida St. 27B, 50-375 Wroclaw, Poland; jaroslaw.kroliczewski@upwr.edu.pl; 5Department of Environment Hygiene and Animal Welfare, Wroclaw University of Environmental and Life Sciences, ul. Chelmonskiego 38C, 51-631 Wroclaw, Poland; robert.kupczynski@upwr.edu.pl

**Keywords:** cows, rumen fermentation, saponite clay, VFA, methane

## Abstract

**Simple Summary:**

The use of various fermentation modifiers in ruminant diets is one of the best strategies for regulating greenhouse gas emissions, primarily carbon dioxide and methane (CH_4_). Feeding complex mineral substances of natural origin can provide a quick and safe result. Our in vitro studies confirmed these expectations, and saponite clay, due to its high adsorption properties, significantly reduced the production of CH_4_ and other gases.

**Abstract:**

Reducing the emission of global warming gases currently remains one of the strategic tasks. Therefore, the objective of our work was to determine the effect of saponite clay on fermentation in the rumen of cows. The pH, total gas production, CH_4_, and volatile fatty acid (VFA) production in ruminal fluid was determined in vitro. Saponite clay from the Tashkiv deposit (Ukraine) has a high content of silicon, iron, aluminum, and magnesium. The addition of 0.15 and 0.25 g of saponite clay to the incubated mixture did not change the pH but reduced the total production (19% and 31%, respectively) and CH_4_ (24% and 46%, respectively) in the ruminal fluid compared to the control group and had no significant effect on the total VFA levels, but propionic acid increased by 15% and 21% and butyric acid decreased by 39% and 32%, respectively. We observed a decrease in the fermentation rates, with a simultaneous increase in the P:B ratio and an increase in the fermentation efficiency (FE) in the groups fermented with saponite clay, probably a consequence of the high efficiency in the breakdown of starch in the rumen. Therefore, further in vivo studies to determine the effective dose and effect of saponite clay on cow productivity and the reduction of gas emissions are promising and important.

## 1. Introduction

The rapid pace of global warming is increasingly causing public concern. These trends are related to electricity production, goods production, deforestation, transportation use, and food production [1]. According to data from the UN World Meteorological Organization, in 2023, our planet has warmed by 1.4 degrees above the preindustrial 1850–1900 baseline due to greenhouse gas emissions. Furthermore, the latest report published in April 2023 by the Intergovernmental Panel on Climate Change reported that agriculture accounts for 22% of global greenhouse gas emissions, rendering the agricultural sector one of the leading contributors to global warming [2,3]. Different strategies, such as dietary modification, genetic selection, microbiome manipulation, and feed additives, such as plant secondary metabolites, CH_4_ inhibitors, lipids, essential oils, and algae, are used to reduce the enteric emissions of gases like CH_4_ or carbon dioxide during the production of food products [4]. The general importance of minerals in the body is known and indisputable. Therefore, to correct the emission of gases by controlling the activity of methanogens, it can be useful to use various mineral additives of natural origin, which also play the role of adsorbents [5]. Several clays can be found in a multitude of environments that are used as feed additives for improving feed quality and enhancing the nutritive value of animal diets [6].

The most commonly used clay-based mineral additives are bentonite, sepiolite, and natural zeolite. Bentonite is a heterogeneous rock, mostly composed of montmorillonite, a clay aluminum silicate, and has a 2:1-layer structure of tetrahedral and octahedral sheets, with large pores interconnected with small channels [5,7]. It has the capacity to substitute Si^4+^ for Al^3+^ in the tetrahedral sheet and substitute trivalent cations for divalent cations in the octahedral sheet, which is a highly colloidal and flexible clay [7]. Bentonite has been approved for use in lowering the mycotoxin contamination of animal feed [8]. In addition, there are other beneficial applications for this natural clay, such as preventing the decline in rumen pH during starch fermentation [9] and reducing the concentration of ammonia in the rumen [10]. On the other hand, administering bentonite does not seem to change the overall fermentation process in the rumen, but it changes the metabolome and mineral concentrations in the rumen fluid [11,12].

Sepiolites, such as kaolinite and smectite, are a clay mineral that belongs to phyllosilicates with a phyllosilicate structure of 2:1, characterized by their remarkable structural stability, high porosity, and surface area, as well as their strong absorptive power. Sepiolite contains a continuous two-dimensional tetrahedral sheet; however, it lacks continuous octahedral sheets [13]. Chemically, sepiolite is a natural hydrous magnesium silicate whose individual particles have a needle-like morphology that contains micropores and channels. Together with its fine particle size and fibrous structure, sepiolite has a high surface area [14]. However, in contrast to the dietary use of bentonite and zeolite, which have high absorption and ion exchange capacities and interact with minerals, sepiolite has a low cation exchange capacity [15]. Sepiolite also shows a higher capacity for adsorption than bentonite, possibly by interacting with specific components of the cell membrane and selectively adsorbing methanogenic bacteria [16]. Furthermore, sepiolite is used as an additive in animal feed to reduce the rate of passage through the gastrointestinal tract, allowing better digestion and contributing to rumen buffering [17,18]. It was suggested that sepiolite may contribute to rumen buffering; however, the data presented in the literature are contradictory [19,20]. In addition, sepiolite increases the VFA concentration, reduces the number of bacteria, and reduces the production of methane and ammonia in the rumen [21,22]. In contrast to sepiolite and smectite, kaolinite is a dioctahedral 1:1-layer clay mineral containing only trace to minor amounts of iron [23]. In kaolinite, the hydrogen bonds between the neighboring octahedral and tetrahedral layers prevent water absorption, and the mineral only exposes its external surface to the environment [11].

Zeolites, such as clinoptilolite, are a 3-dimensional crystalline structure composed of aluminosilicate (SiO_4_ and AlO_4_) tetrahedra joined with oxygen atoms to form large pores that form larger molecular sieves [7].

They have the ability to lose and gain water reversibly and have a higher capacity to exchange cations (such as K^+^, NH_4_^+^, Ca^2+^, Na^+^, and Mg2+) than bentonite clay, as well as altering the rumen retention time while improving nitrogen utilization in ruminants [24,25,26]. Furthermore, zeolite can improve ruminal digestion but does not appreciably affect net ruminal microbial growth or nitrogen utilization, as well as alters the molar proportions of the ruminal VFAs (decreasing the acetate: propionate ratio) [24]. Previous studies have shown a positive, negative, or non-existent effect of zeolite addition on rumen fermentation, especially ruminal NH3, and pH [25,27,28]. This could be due to the type of protein rations, urea supplementation, feed level of neutral detergent fiber (NDF), type of animal, roughage-to-concentration ratio, etc. [29].

Saponite clay is interesting among the clay minerals because it contains different macro, micro, and ultra-microelements and also has significant ion capacity and sorption properties [30]. These applications depend on properties, such as size, cation exchange capacity, plasticity, catalytic activity, swelling behavior, permeability, and substitutions in the saponite structure [31]. It was established that raw saponite clay from the Tashkiv deposit (Ukraine) consists mainly of trioctahedral saponite with an admixture of dioctahedral nontronite and associated minerals (quartz, hematite, and anatase). Raw saponite clay is characterized by a high content of iron (19.3%) and titanium (1.1%). Iron is present as hematite particles or as a cation in the interlayer space [32].

Research by Zarate-Reyes et al. [33] established an effective suppression of antibiotic-resistant gram-negative bacteria growth when using iron saponite with impurities of quartz, feldspar, and calcite. According to their data, labile or bioavailable iron ions in clay induced quantitative -HO production as well as oxidative stress. Basargin et al. [34] described the effectiveness of using saponite clay to improve the biological value of pig meat. Furthermore, Savchuk et al. [35] evaluated that the feeding of mineral supplements based on saponite and glauconite increased the milk productivity of first-born cows by 7.9–15.1%. At the same time, feeding saponite in combination with selenite to cows during the dry period and after calving normalized metabolic processes and increased milk productivity. An effective method for the prevention of gastrointestinal diseases in calves and pathological conditions during and after calving in cows is the use of saponite and saponite plus selenite. Such a mineral supplement also helps to increase the average daily milk yield [36]. According to the manufacturer’s instructions, saponite clay is recommended as a feed additive/sorbent for continuous feeding without a time limit due to its ability to adsorb and remove substances dangerous to the body while provoking the absence of toxic effects [37].

The mitigation of ruminant CH_4_ production remains a formidable challenge for both improving feed conversion efficiency and decreasing the emissions of this highly potent greenhouse gas. The majority of research on clay has focused on mineral supplementation and against toxins, and a few studies have been performed on the effect of clay supplementation on CH_4_ production in ruminants. In one of them, Váradyová et al. [38] showed that the dolomites from tested natural sources can be principally used to decrease CH_4_ production without effects on the VFAs. Furthermore, in a series of preliminary in vitro studies, it was discovered that certain clays could significantly, albeit inconsistently, reduce CH_4_ production. The most reliable outcomes were obtained with kaolinite when the initial pH of the inoculum, which consisted of bovine rumen, was within the range of 6.0 to 6.2 [39].

Therefore, for the first time, we studied the effect of saponite clay on ruminal fermentation parameters during in vitro studies, including the production of gases, pH level, and VFA profile.

## 2. Materials and Methods

### 2.1. Animals

The studies were carried out at the Research and Teaching Station of the University of Environmental and Life Sciences in Wrocław, Poland. The ruminal fluid was collected through cannulas from non-lactating Polish Holstein-Friesian cows of the black and white variety, approximately 610 ± 26 kg of BW (multiparous cows), at 3.5 ± 0.25 BCS. The animals were kept in a captive system and fed a standard balanced diet according to INRA, Feeding Recommendations for Ruminants [40]. The dry matter intake (DMI) was 11 kg/day. Fresh feed was delivered twice a day (from 6 to 7 a.m. and 2 to 3 p.m.) During the entire trial, the offered feed was adjusted daily to target a refusal rate of 5 to 10%.

The composition of the food ration was as follows: grass silage 69.50% DM, meadow hay 11.90% DM, barley ground 9.53% DM, rapeseed meal 7.91% DM, fodder chalk (the raw mineral material contained up to 98% CaCO_3_ (usually 91–93%), and from 0.11 to 0.64% Fe_2_O_3_ and trace amounts of macro and micro-elements, such as Mg, K, Na, P, Cu, Zn, and Mn), 0.174% DM, premix 0.986% DM.

The rumen contents were collected through the fistula from rumen-cannulated cows in an amount of 800 mL before the morning feeding and were placed in thermoses with an internal temperature of 39 °C. They were also coated with parafilm to avoid undue exposure to air and transported with care in a thermal bag to the laboratory in 20 min. Before use, the rumen fluid was filtered through a four-layer cheesecloth. At one time, the content was retrieved from two cows alternately in weekly intervals. Seven biological samples of rumen fluid were collected in 4 weeks.

The experimental procedures were carried out according to the European Union Directive 2010/63/EU on 22 September 2010, which entered into full effect on 1 January 2013 [41]. The study did not involve procedures that caused pain, discomfort, or distress to the animals and was reviewed by the local ethics committee on animal experiments (protocol No. 053/2019).

### 2.2. Saponite Clay

The saponite clay was a loose, dry powder (fraction: 0–0.2 mm) of brown color without extraneous odors or tastes (Figure 1). The mining site was the village of Ulashanivka, Shepetivskyi district, Khmelnytskyi region, Ukraine. The excavation depth was 9 m. The official producer and rights holder of the production facilities was NATURAL MINERALS LLC. On the market, saponite clay is usually sold under the trade name SAPOKORM^TM^. According to the official guidelines, the theoretical formula of saponite was M^+^_y − x_[(Mg,Fe^2+^)_3 − x_(Al,Fe^3+^)_x_] ^+ x^[(Si_4−y_Al_y_) ^− y^O_10_]·(OH)_2_·nH_2_O.

### 2.3. Chemical Analysis

In representative samples of the corn and grass mixture, the basic nutrients were measured according to the standards of the association of official analytical chemists (AOAC) as follows: dry matter (AOAC method 934.01) [42], Kjeldahl N (984.13 of 143 AOAC 2005) using a Kjeltec 2300 Foss Tecator apparatus (Häganäs, Sweden) for the calculation of the crude protein (CP) as Kjeldahl N × 6.25 [42]. The ether extracts (EEs) were determined according to AOAC 2005 method 920.39 using a Fibertec Tecator apparatus (Häganäs, Sweden). AOAC method 942.05 was used to determine the crude ash [42,43]. The crude fiber (CF) was measured using the AOAC method 978.10. The neutral detergent fiber and acid detergent fiber (ADF) fractions were measured using a Fibertec Tecator apparatus (Häganäs, Sweden) according to Holst and AOAC method 973.18, respectively [44]. The NDF and ADF were expressed exclusively in the ash. The non-structural carbohydrate (NSC) content was calculated according to the National Research Council guidelines as follows: 100 − (Ash + CP + EE + NDF), with the ash, CP, EE, CF, and NDF contents expressed as the % of DM.

The determination of the trace element content (Table 1) in the feed mixture and saponite clay was carried out using SEO-SEM Inspect S50-B (SELMI, Sumy, Ukraine) using an AZtecOne energy dispersive spectrometer AZtecOne with an X-MaxN20 detector (manufactured by Oxford Instruments plc.) according to the method described by Casu et al. [45]. The mass fractions of the elements in the field of view were determined via X-ray microanalysis (EDX) on the energy values of the characteristic X-ray peaks of the chemical elements in the Be-U range.

### 2.4. Fermentation in the In Vitro Studies

The fermentations were carried out using the Ankom RF Gas Production System, (ANKOM Technology, Macedon, NY, USA) consisting of glass bottles equipped with temperature (5–60 °C) and pressure sensors (pressure range: from −69 to +3447 kPa, resolution: 0.27 kPa, accuracy 0.1%) under anaerobic conditions at a temperature of 39 °C [46,47,48,49].

Seven biological samples of rumen fluid were collected and prepared for the fermentation experiments. Each biological sample was homogenized and mixed with prewarmed solution in a 39 °C buffer [50] in a 1: 3 ratio (20 mL of rumen fluid to 60 mL of buffer) and transferred to prewarmed glass bottles. Then, each biological sample was divided into three groups: group C (control), which received 1 g of a mixture of corn and grass feed (1/1); group I, which received 1 g of a mixture of corn and grass feed (1/1) supplemented with 0.15 g of saponite clay; and group II, which received 1 g of a mixture of corn and grass feed (1/1) supplemented with 0.25 g of saponite clay. To determine the fermentation parameters, three technical replicates for each group per biological sample were carried out using the Ankom system. Therefore, there were seven biological replicates per treatment group, followed by three technical replicates for each group (*n* = 21). In total, we analyzed 63 samples in our studies.

The bottles were tightly closed and then vented with CO_2_ through a Luer port to achieve anaerobic conditions until the internal pressure exceeded 8 psi. Once the gases were released from the bottle, the real-time automatic recording of gas production was initiated and measured with the following parameters: a recording interval of 600 s, a threshold of 1.5 psi for the automatic release of the accumulated gases to prevent CO_2_ transfer in the medium at a high gas pressure [46,47,48], a valve opening time of 250 ms, and a mixing interval of 50 c.p.m. During fermentation, the bottle was kept in shaking water baths at 39 °C to ensure that the solution was at a stable temperature during the measurements. After 24 h of incubation, the cumulative gas pressures were read from the data set from the Ankom RF instrument [48]. Using the “ideal” gas law equation, n = p(V/RT), the measured gas pressure was converted to moles of gas produced, with the following factors [47]:

n = gas produced in moles (mol);

p = pressure in kilopascals (kPa);

V = headspace volume in the glass bottle in liters (L);

T = temperature in Kelvin (K);

R = gas constant (8.314472 L·kPa/K/mol).
Gas produced in mL = total gas production (mol) × 22.4 (molar volume) × 1000

At the end of the fermentation process, the pH was measured in the liquid content using the inoLab^®^ pH 7110 SET 2 (Karlsruhe, Germany) pH meter with a SenTix 41 electrode and a temperature sensor. Then, to stop the fermentation process, formic acid (0.1 mL/2 mL of solution) was added.

The methane level was analyzed after incubation using a gas chromatography method. The gas was collected for the CH_4_ analysis using a 10 mL gas-tight syringe (Agilent Technologies, Santa Clara, CA, USA) from a closed bottle through a side port after the fermentation was complete. During the sampling, the syringe was flushed with the produced gas to ensure that a homogeneous sample was collected. The samples were injected directly into the column for analysis using a gas chromatography method (Agilent Technologies 7890A GC System, Santa Clara, CA, USA equipped with a thermal conductivity detector (TDC), a flame ionization detector (FID), and two columns: Porapak Q and HayeSep Q (Supelco, Bellefonte, PA, USA), with a 5A molecular sieve (flow: 25 mL/min)) with four replicates per treatment [51].

To analyze the experimental chromatograms, a comparative analysis of the retention times of the CH_4_ standards produced by the company Linde Group was performed using ChemStation software no. B.03.02 (Agilent Technologies, Santa Clara, CA, USA).

The liquid contents of the samples were analyzed to determine the overall concentration of the VFAs and the percentage of individual acids using a gas chromatograph analyzed on a 7890A gas chromatograph (Agilent Technologies, Santa Clara, CA, USA) with a flame ionization detector equipped with an Agilent J&W DB-WAX column, with helium as the carrier gas (flow: 25 mL/min). The formula provided below was used to calculate the CH_4_ yield [52]:$${\mathrm{CH}}_{4}\;\mathrm{mL}=\frac{percentage\;concentration\;of\;{CH}_{4}\times gas\;produced\;in\;mL}{100}$$

Based on the analysis of the level and profile of the VFAs, the VFA utilization index (NGR) of the volatile fatty acids (NGR) expressed as the ratio of nonglucogenic to glucogenic VFAs was calculated according to the formula NGR = (A + 2B + Bc)/(P + Bc) [53,54]. The fermentation efficiency coefficient was calculated according to the formula FE = (0.622A + 1.092P + 1.56B) 100/(A + P + 2B) [55] where A, P, and B are the percentage contents in the VFA total concentration of the tested (mol%) acetic acid, propionic acid, and butyric acid, respectively, and Bc is the percentage of the total concentration of VFA: valeric acid and branched short-chain fatty acids. Additionally, the index of cell yield (CY) of the mixed ruminal microorganisms was calculated using the equation developed by Chalupa [56]: CY = (A + P + B + V) × 0.03, where A, P, B, and V are the concentrations (mmol/L of ruminal fluid) of acetic, propionic, butyric, and valeric acid, respectively. The CY parameter was calculated according to a calculation published by Chalupa [56] based on the value of 30 g of bacteria formed per mole of VFAs, produced and expressed in g/L.

Additionally, the acetate: propionate (A:P) and propionate: butyrate (P:B) ratios were calculated. Furthermore, the hydrogen recovery (H_2_%) and the ratio of hydrogen consumption via CH_4_/VFAs were estimated using the following formulas [57]:$$H_{2}\;(\%)=\frac{(4M+2P+2B)\times 100\%}{\left(2A+P+4B\right)}\;{\mathrm{CH}}_{4}/\mathrm{VFA}=\frac{4M}{\left(2P+2B\right)}$$
where M = CH_4_ [mmol/L], P = propionic acid [mmol/L], and B = butyric acid [mmol/L] in the fermenting fluid.

### 2.5. Statistical Analysis

The results were subjected to statistical analysis using one-way analysis of variance (ANOVA) in the Statistica 13.3 program [58]. The analysis was preceded by evaluating all the data for normality using the ANOVA procedure. The significance at *p* < 0.05 of the differences between the groups at *p* < 0.05 was determined using Duncan’s multiple comparison test. The analysis data represent seven biological replicates with three analytical repeats for each biological sample (n = 21).
The statistical model was as follows: Zij = x + fi + eij
where Zij represents the value of each individual observation (an individual bottle was the experimental unit), x is the overall mean, fi is the fixed effect of the treatment (different additives; C, I, and II), and eij is the residual error.

## 3. Results

The elemental compositions of the saponite clay and feed mixture samples used in this study are shown in Table 1. Oxygen was the main element in the analyzed saponite clay sample, followed by silicon, iron, carbon, aluminum, and magnesium. At the same time, carbon was the main element of the feed mixture, followed by oxygen and potassium. In contrast to the feed mixture, iron and aluminum, as well as a small percentage of titanium, were found in the composition of saponite clay. On the other hand, saponite lacks sulfur, phosphorus, and chlorine. The total content of the main trace elements (calcium, magnesium, sulfur, phosphorus, chlorine, and silicon), except potassium—3.53%, in the feed mixture was quite low—2.86%. In turn, the saponite clay contained less than 1% of potassium and more than 10% of iron. No significant effect of saponite clay was observed on the pH of the fermentation fluid content after 24 h of fermentation (Table 2). However, the pH was stabilized at pH ~6.5 as opposed to the values in the control group. Group II showed significantly lower levels of hydrogen recovery (H_2_%) and CH_4_/VFA (*p* < 0.05), as indicated in Table 2. Following a fermentation period of 4 h, group II exhibited a lower level of gas production compared with the control group. In addition, after 24 h, the utilization of saponite clay caused a modification in the overall generation of gases and CH_4_. Group II exhibited a noteworthy decrease (*p* < 0.05) in the generation of gases in comparison to the other groups (Table 3).

The addition of saponite clay to the in vitro fermentation contents did not affect the level of total VFA production as well as the amounts of acetic acid, valeric acid, hexanoic acid, and heptanoic acid (Table 4). The use of saponite clay in the fermented fluid in groups I and II resulted in a higher proportion (*p* < 0.05) of propionic acid compared to group C. However, the change in butyric acid production was the opposite; the addition of saponite clay in groups I and II caused a decrease (*p* < 0.05) in butyric acid production compared to group C. The level of branched volatile fatty acid production, including isobutyric acid and isovaleric acid, remained unaffected by the utilized substrates. There was only a higher but not significant level of isocaproic acid in group C compared to the other groups.

The fermentation efficiency increased (*p* < 0.05) in group II compared to group C. In the fermentation fluid, regardless of the clay level used (groups I and II), the A:P ratio and the NGR were lower (*p* < 0.05) compared to group C. However, the P:B ratio in group C was lower (*p* < 0.05) in comparison to groups I and II.

Regarding the VFA concentration, the CY index was calculated in our study. The VFA production rate is dependent on the availability and composition of the substrate, the rate of depolymerization, and the microbial species present [59]. We observed a significantly lower value of CY in the groups treated with saponite.

## 4. Discussion

The digestion process in ruminants produces gases as a byproduct of metabolism. The limitation of gas emissions and, in particular, CH_4_ production is highly desirable [4]. However, reducing CH_4_ emissions can negatively affect the efficiency of feed use efficiency [59,60,61]. Animal nutrition determines the changes in the rate of methanogens and H_2_ in cow rumen, which, in turn, affects the level and profile of VFA production and CH_4_ synthesis [62]. Lower CH_4_ emissions in conjunction with a stable VFA concentration during rumen fermentation after the application of the feed additive can be interpreted as a positive development [63].

Breakthroughs in dairy nutrition and health have focused on the prevention of acute and subacute ruminal acidosis. Moreover, ruminal pH is a fermentation parameter that determines rumen alkalinity. Therefore, pH is a risk factor for rumen health, microbial growth, microbial change, and stability, as well as cellulose digestion, biohydrogenation, methanogenesis, defaunation, and the rate at which volatile fatty acids (VFAs) are absorbed [63].

The pH status could predict the type of diet provided to the animals, and the degree of rise or decrease could predict the rate of fermentation [64,65]. Variations in ruminal pH cause changes in the bacterial population [66,67]. Some clays, such as bentonite, also known as smectite clay, can stabilize the pH of the rumen. This owes to the excellent clay structure in which ions can attach and the high “swelling” qualities. Saponite clay, such as bentonite clay, is a three-dimensional smectite group mineral. Because of the way the space is organized, it has a changeable net negative charge that is balanced by adsorbed Na, Ca, Mg, Al, and H on the surfaces between the layers. Furthermore, saponites also have medium to high cation exchange capacity and exhibit remarkable hydration properties and the ability to dramatically influence liquid flow due to interlamellar surfaces [68]. Rindsig et al. [69] tested nursing cows at risk for acute and subacute ruminal acidosis with 5% and 10% bentonite clay. The milk fat production increased in cows who were fed clay versus those not fed clay, implying that the ruminal acetate and propionate concentrations were more effectively balanced in their rumen and blood.

Sulzberger et al. [70] evaluated the effectiveness of a clay product on cows. They discovered that clay-fed cows had higher rumen pH. Clays also modulate the ruminal microbiome by regulating the availability of ammonia since these minerals can reversibly absorb this molecule as a function of its concentration, acting as an ammonia buffering system for the ruminal microbiome [70]. Furthermore, clay increased the rumen and fecal pH, decreased the amount of time that the rumen pH spent below 5.6, and ultimately increased the milk fat yields that were shown to decrease in SARA cases [69]. Understanding how rumen fermentation is affected by saponite clay deserves attention because the pH reflects the balance between the production of organic acids and their absorption and neutralization in the rumen [71].

Our study has shown that the addition of saponite clay stabilized the higher pH that occurred during fermentation, which is the desired result. An increase in ruminal pH was also observed with the addition of zeolite clays in other studies with high-concentration diets [72,73]. Bentonite had the same effect on pH as zeolite, although to a smaller extent, indicating that these minerals can help to maintain a balanced pH in the rumen, particularly during starch fermentation, as reported by Fisher and Mackay [9,69].

It is well known that clay as zeolites can change the fermentation patterns in the rumen, which in turn changes the molar proportions of VFAs. This is true even though different studies have found different pH levels in the rumen [73]. Nevertheless, the fluctuations in the relative abundance of VFAs are not consistent. Mineral clays were found to increase the proportion of acetate [9] and decrease the proportion of propionate and valerate [74] in some studies while increasing the proportion of propionate in others [75]. Furthermore, in cow feeding, the supplementation of feed with an additional source of calcium and magnesium slightly affected the level and profile of VFAs [76,77,78]. The addition of calcium-magnesium salt reduced the level of butyrate acid in the rumen content [77]. The addition of iron to the food resulted in a reduction in the levels of VFAs and acetic, propionic, and butyric acids in the rumen of sheep [79]. In the present investigation, there were no changes in the VFA level, and the proportion of butyric acid was reduced in the groups with the addition of saponite clay. This may be because the saponite clay had a much higher level of calcium and magnesium compared to the feed used. It is assumed that the increase in the iron level in the clay did not affect the fermentation profile. Damato et al. [11] discovered that the use of bentonite clay during in vitro fermentation using ruminal fluid from dairy cows increased propionic acid and decreased butyric acid, which is consistent with our findings.

In our investigation, a significant reduction in CH_4_ emissions was observed in the fermented group with the addition of saponite clay, while there were no changes in the level of total VFA production. The in vitro study by El-Nile et al. [80] showed a link between the addition of nanozeolite and lower levels of truly degraded organic matter, gas production, and CH_4_. This suggests that mineral clay might be able to improve feed breakdown while changing the rumen fermentation patterns to produce less CH_4_.

The hydrogen produced during the fermentation process can be transferred from the methanogenesis pathway to the propionic acid production pathway [4]. In this study, the use of saponite clay resulted in a reduction in the total gas and CH_4_ production in group II and increased the proportion of propionic acid. Moreover, the CH_4_/VFA ratio was also reduced after the addition of clay to the fermenting fluid. This could suggest that the inhibition of methanogenesis may induce the accumulation of unused hydrogen that should be used to produce VFAs. As a result, it was suggested that a decreased H_2_% level might be a sign of a hydrogen transfer to VFAs [81]. In addition, H_2_% is positively correlated with the rate of methane production during rumen fermentation [82].

Furthermore, the results of this research demonstrated that saponite clay reduced CH_4_ production in fermentation fluid. The SEO-SEM and EDX analyses showed that saponite clay contains 10.94% iron and 6.08% aluminum. The inhibitory effect of released aluminum from kaolinite on methanogenesis was previously described and showed that a stronger inhibitory effect was observed with a higher aluminum concentration [23]. Furthermore, Liu et al. [23] showed that even 0.1 mM aluminum (~30 μM dissolved aluminum) can inhibit the growth of Methanosarcina mazei and Methanothermobacter thermautotrophicus. This effect might have been the reason for the incomplete iron reduction since hydrogenotrophic methanogens should easily be able to transform to Fe(III) in their amorphous form [83]. In our study, where the concentration of aluminum in clay added to fermentation fluid was between 0.3 and 0.5 mM, we also observed a significant reduction in the number of microorganisms expressed by the CY index. Previous studies using a variety of microorganisms (excluding methanogens) reported similar findings, such as decreased microbial activity after the addition of phyllosilicates [84,85]. Further possibilities of saponites’ action in limiting methanogenesis may be related to iron reduction by methanogens. This suggests that iron bioreduction itself may have caused the suppression of the initial methanogenesis and could be explained by the fact that methanogens can divert most electrons from methanogenesis to Fe^3+^ reduction when methanogens are in an environment of iron-containing clay minerals [86,87]. However, this mechanism requires further investigation.

The A:P ratio and NGR index are two important indicators that can determine the functioning of the rumen [80]. The NGR parameter is extremely highly positively correlated with the intensity of CH_4_ production [66,88,89]. This study has shown that the NGR index was related to methanogenesis. The NGR reduced from 4.87 to 3.58, while the production of methane decreased from 33.05 mL to 17.75 mL/L. A reduction in the total gas production of 65.85 mL/L was also observed.

FE is another important indicator of the fermentation process that takes place in the rumen, which is calculated using VFAs. Increasing the FE combined with reduced CH_4_ production is also desirable [90]. In our studies, the FE increased from 73.76% to 74.83%, while the CH_4_ emissions were reduced by 15.3 mL/L of fermenting fluid throughout 24 h.

Increasing the proportion of easily fermentable carbohydrates, including starch, in the diet often results in faster fermentation, an increase in the molar fraction of propionate, and a lower proportion of acetic acid and NGR [59,91]. When saponite clay was added to the fermenting fluid, the values of the fermentation indicators, such as the NGR and A:P, decreased, while the P:B ratio and FE simultaneously increased. This may result due to the high effective degradability of starch in the rumen, which was probably stimulated by the saponite clay. Furthermore, CH_4_ production is part of the rumen fermentation process, where CH_4_ is synthesized by methanogen microorganisms. The use of hydrogen by microorganisms can lead to two other reactions, resulting in the synthesis of acetate or propionate [4]. In our experiment, the A:P ratio favored the production of propionate. It is possible that the saponite clay stopped the activity of the microorganisms that produce CH_4_ or steered the fermentation process toward the propionate path. Methanogenic bacteria produce CH_4,_ and methanotrophic microorganisms use it up. This balance is what allows biological CH_4_ production to be possible. It is conceivable that clay additives might be able to cause an imbalance in favor of methanotrophic microbial activity. Of course, we cannot rule out that the saponite clay decreased the digestion of the organic matter, thus reducing the total VFA production, total gas production, and methane production [6,92]. However, we must bear in mind that the total VFA concentrations in groups I and II were lower than in the control group, but not significantly.

## 5. Conclusions

The results show a positive effect of saponite clay on the fermentation profile in the rumen of cows. The addition of saponite clay reduced methane production while maintaining constant VFA production, indicating that stable processes are taking place in the rumen fluid. Saponite clay increased the fermentation efficiency and propionic acid. Furthermore, the additive stabilized the fermentation processes that occur in the rumen fluid. In the future, further in vivo research should be conducted to determine the impact of using saponite clay as a feed additive for cows to increase the production and quality of animal products and reduce gas emissions, including CH_4_. Regarding the limits of this study, we did not perform a detailed analysis of the microorganisms before and after fermentation, nor an analysis of the effect of saponite clay on the microorganisms using the in vitro study. The latter would address the questions of which clay properties affect the microorganisms more and which of them, considering not only bacteria—including methanogenic ones—but also protozoa, and whether or not the metal ions that clay contains could play a fundamental role. We believe that in vivo studies should be carried out only after such analyses.

## Figures and Tables

**Figure 1 animals-14-00738-f001:**
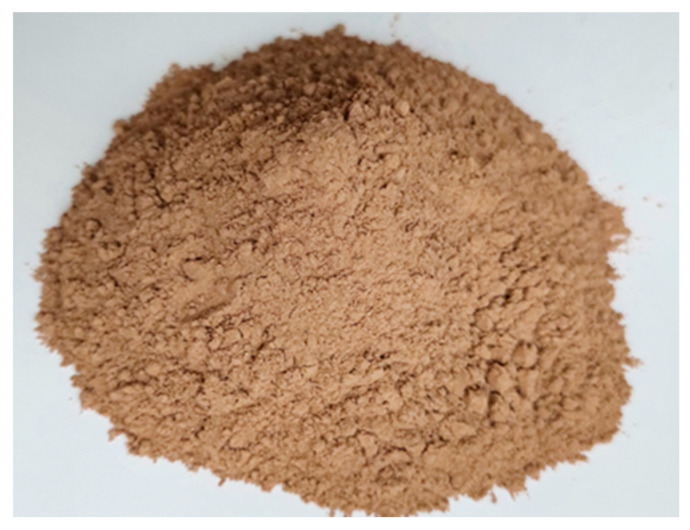
Saponite clay.

**Table 1 animals-14-00738-t001:** Chemical composition of the incubated substrates.

Item	Unit	Feed Mixture	Saponite Clay
Dry matter (DM)	g/kg	555.25	-
Crude protein	g/kg DM	129.20	-
Ether extract	g/kg DM	23.85	-
Crude fiber	g/kg DM	98,63	-
Neutral detergent fiber	g/kg DM	556	-
Acid detergent fiber	g/kg5 DM	254	-
Crude ash	g/kg DM	50.20	-
Non-structural carbohydrates	g/kg DM	240.75	-
Gross energy	MJ/kg DM	17.50	-
K	%	3.53	0.90
Ca	%	0.28	1.97
Mg	%	0.43	4.98
S	%	0.67	-
P	%	0.54	-
Cl	%	0.61	-
Si	%	0.33	15.81
Fe	%	-	10.94
Al	%	-	6.08
Ti	%	-	0.15
O	%	28.37	52.03
C	%	65.19	7.10

**Table 2 animals-14-00738-t002:** The influence of saponite clay on the pH, hydrogen recovery (H_2_ %), and ratio of hydrogen consumed via CH_4_/VFA in the fermentation fluid.

	Group			SEM	*p*-Value
	C	I	II		
pH	6.32	6.48	6.49	0.055	0.344
H_2_ (%)	64.39 ^a^	65.28 ^a^	56.59 ^b^	2.268	0.034
CH_4_/VFA	1.09 ^a^	1.12 ^a^	0.75 ^b^	0.092	0.046

SEM, standard error of the mean; ^a,b^ values differ significantly between the groups (*p* < 0.05).

**Table 3 animals-14-00738-t003:** The profile of total gas production and the cumulative production of CH_4_ in the investigated treatments.

	Group			SEM	*p*-Value
Incubation Time (h)	C	I	II		
	Total gas production [mL/L] *				
4	14.35 ^a^	5.10 ^b^	4.15 ^b^	0.275	0.025
8	59.05	43.05	39.95	8.151	0.159
12	101.50	87.15	76.30	6.235	0.243
16	133.50	112.90	101.80	7.685	1.940
20	167.05	141.00	125.05	9.615	0.234
24	212.05 ^a^	171.50 ^ab^	146.20 ^b^	0.905	0.029
	Methane [mL/L]				
24	33.05 ^a^	25.95 ^ab^	17.75 ^b^	0.105	0.014

* Cumulative over the time measurements; SEM, standard error of the mean; ^a,b^ values differ significantly between the groups (*p* < 0.05).

**Table 4 animals-14-00738-t004:** The influence of saponite clay on VFA production in the fermentation fluid.

	Group			SEM	*p*-Value
	C	I	II		
Total VFA [mmol/L]	87.90	69.20	69.61	3.969	0.078
	Individual VFA, [mol/100 mol]				
Acetic acid	66.55	68.33	66.38	0.505	0.274
Propionic acid	18.85 ^b^	21.66 ^a^	22.89 ^a^	0.593	0.006
Isobutyric acid	0.48	0.59	0.59	0.049	0.563
Butyric acid	11.91 ^a^	7.23 ^b^	8.13 ^b^	0.740	0.017
Isovaleric acid	0.68	0.74	0.64	0.032	0.517
Valeric acid	0.96	0.91	0.82	0.056	0.567
Isocaproic acid	0.15	0.00	0.04	0.026	0.064
Hexanoic acid	0.13	0.20	0.01	0.049	0.315
Heptanoic acid	0.31	0.34	0.49	0.057	0.353
FE (%)	73.76 ^b^	74.13 ^ab^	74.82 ^a^	0.180	0.029
A:P	3.60 ^a^	3.16 ^b^	2.93 ^b^	0.097	0.006
P:B	1.91 ^b^	3.01 ^a^	2.82 ^a^	0.149	0.029
NGR	4.87 ^a^	3.75 ^b^	3.58 ^b^	0.198	0.006
CY ^1^	7.46 ^a^	4.49 ^c^	5.88 ^b^	0.587	0.002

SEM, standard error of the mean; ^1^ g/L of undiluted ruminal fluid; ^a,b^ values differ significantly between the groups (*p* < 0.05), FE, fermentation efficiency; A:P, ratio of acetic acid to propionic acid, P:B, ratio of propionic acid to butyric acid; NGR, VFA utilization index; CY, cell yield index.

## Data Availability

Data are contained within the article.

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
