# Peer review of "The Effect of Saponite Clay on Ruminal Fermentation Parameters during In Vitro Studies"

_animals, 2024, doi:10.3390/ani14050738_

Round 1

Reviewer 1 Report

Comments and Suggestions for Authors

Introduction:
The primary concern is that the authors should propose their own hypothesis in this study. For example, how does saponite clay reduce methane emissions? Is it through physical adsorption or through biological pathways such as rumen microorganisms? If it is through physical adsorption, the Introduction should clarify the properties, effects, and results of methane adsorbents currently available. A comparison with saponite clay should also be included.

Materials and Methods:
1. Key information, such as the age and parity of the cows, as well as the timing of rumen fluid collection, has not been provided.

Results:
1. I believe Table 1 and Figure 2 should be included in the Materials and Methods section since they represent basic properties of the treated materials that need to be explained in detail in the Results, rather than presented as independent results.
2. Some abbreviations, such as A:P share, NGR ratio, P:B ratio, etc., need to be explained.
3. Considering the rumination process in cows, if saponite clay reduces methane emissions through adsorption, the effects on methane release at different incubation time points should be provided, rather than just a single time point. If the adsorption rate is slow or the adsorption capacity is weak, methane could still be released through the oral cavity or anus.
4. The authors need to provide results on rumen microorganism analysis to determine the potential impact of saponite clay on methane-producing microorganisms and ascertain whether its effect on methane reduction is through physical adsorption or biological activity.
5. The authors should supplement with in vivo experiments, such as respiratory gas analysis, to confirm the actual effects. In vitro experiments alone are insufficient to demonstrate practical results.

Discussion:
The authors have primarily discussed saponite clay as a typical bentonite. A more in-depth discussion is needed, such as highlighting the unique characteristics of saponite clay compared to other bentonites. Does it possess superior methane reduction capabilities compared to other bentonites? What are its distinctive features compared to other types of methane adsorbents?

Comments on the Quality of English Language

English is OK.

Author Response

The Authors would like to thank for insightful comments that will improve the quality of our manuscript. We have followed the Reviewer’s suggestions.

Q: The primary concern is that the authors should propose their own hypothesis in this study. For example, how does saponite clay reduce methane emissions? Is it through physical adsorption or through biological pathways such as rumen microorganisms? If it is through physical adsorption, the Introduction should clarify the properties, effects, and results of methane adsorbents currently available. A comparison with saponite clay should also be included.

R: The introduction has been corrected to take into account sometimes contradictory suggestions from the reviewers and the editor.

Q: Key information, such as the age and parity of the cows, as well as the timing of rumen fluid collection, has not been provided.

R: Suggested data has been added to the publication

Q: believe Table 1 and Figure 2 should be included in the Materials and Methods section since they represent basic properties of the treated materials that need to be explained in detail in the Results, rather than presented as independent results.

R: As suggested by other reviewers, Figure 2 has been removed, while Table 1 has been moved to the M-M section.

Q: Some abbreviations, such as A:P share, NGR ratio, P:B ratio, etc., need to be explained.

R: The shortcuts have been explained

Q: Considering the rumination process in cows, if saponite clay reduces methane emissions through adsorption, the effects on methane release at different incubation time points should be provided, rather than just a single time point. If the adsorption rate is slow or the adsorption capacity is weak, methane could still be released through the oral cavity or anus.

R: It was found that hydrate might form from CH4 gas and water that are absorbed between the molecular plates of montmorillonite (~2%). Moreover, some studies have indicated that clays have a limited effect on the gas adsorption capacity. However, these results are from chemical tests at pressures exceeding those found in the rumen. On the other hand, moisture can directly reduce the adsorption capacity of clay pores because water molecules can reduce the number of potential adsorption sites by blocking micropores or directly occupying polar adsorption sites. Data on the possible methane binding efficiency of clay in animal rumen would be interesting, however, was not a focus of this study. The potential mechanism of action of clay has been discussed in the discussion.

  1. The authors need to provide results on rumen microorganism analysis to determine the potential impact of saponite clay on methane-producing microorganisms and ascertain whether its effect on methane reduction is through physical adsorption or biological activity.

R: Thank you for this suggestion and we agree, that the analysis of microorganism diversity in rumen fluid is the next logical step in this experiment. However, this manuscript is only a part of a large study. Therefore, in this paper, we focused mostly on VFA and gas emission production as this was the main goal. The analysis of microbial content and an in-depth analysis of microbial diversity will be the focus of the next manuscript.

  1. The authors should supplement with in vivo experiments, such as respiratory gas analysis, to confirm the actual effects. In vitro experiments alone are insufficient to demonstrate practical results.

R: Due to the labour and ethical problems indicated by the Ethics committee posed by the use of fistulated and cannulated animals to examine the digestive tract, we used the in vitro digestion method to simulate ruminal fermentation. Rumen fistulation and cannulation are indeed an essential tool for the progress of ruminant research regarding the study of new food sources, particularly in the evaluation of their productivity, health status, or the greater or lesser potential for greenhouse gas production. However, it must be carried out with few animals and subjected to strict clinical and management controls that guarantee their welfare at all times. We agree with the reviewer that the use of in vitro fermenters does not replace the data provided by the live animal but can give repeatable additional information about the changes that take place in the rumen environment under standard conditions, independently of the animal. Moreover, in vitro research is accepted by the Ethics Committee on Animal Experiments. In our case, we only obtained permission to collect rumen contents (protocol No. 053/2019). Therefore, after the conclusion we add information about research limitations.

Q: The authors have primarily discussed saponite clay as a typical bentonite. A more in-depth discussion is needed, such as highlighting the unique characteristics of saponite clay compared to other bentonites. Does it possess superior methane reduction capabilities compared to other bentonites? What are its distinctive features compared to other types of?

R: A comparison of mineral clays has been added to the Introduction section as was suggested by other reviewers. It was found that hydrate might form from CH4 gas and water that are absorbed between the molecular plates of montmorillonite (~2%). Moreover, some studies have indicated that clays have a limited effect on the gas adsorption capacity. However, these results are from chemical tests at pressures exceeding those found in the rumen. On the other hand, moisture can directly reduce the adsorption capacity of clay pores because water molecules can reduce the number of potential adsorption sites by blocking micropores or directly occupying polar adsorption sites. Data on the possible methane binding efficiency of clay in animal rumen would be interesting, however, was not a focus of this study.

Reviewer 2 Report

Comments and Suggestions for Authors

Although the paper has been well prepared, the authors need to produce/improve the following suggestions. 

Authors need to clarify the novelty of the study and this must be clarified in the introdcution seciton. The introduction could benefit from providing a clearer context for the importance of reducing global warming gases and how saponite clay may contribute to this goal.

does the authors are satisfied with the amount of results they have presented or need more research or to combine the in vivo study with these results also.

Specify the type of fermentation being discussed and provide more details on the experimental setup, including the incubated mixture and conditions used in the study.

Ensure that the reported results are presented with precision. Instead of saying "a decrease in fermentation rates," provide specific numerical values or percentages for better clarity.

Be consistent in the terminology used. For example, "saponite clay" is referred to as "saponit clay" at one point.

By addressing these points, the paper can become more informative, clear, and precise, contributing to a better understanding of the study's objectives and outcomes.

Author Response

The Authors would like to thank for insightful comments that will improve the quality of our manuscript. We have followed the Reviewer’s suggestions.

Q: Authors need to clarify the novelty of the study and this must be clarified in the introduction seciton. The introduction could benefit from providing a clearer context for the importance of reducing global warming gases and how saponite clay may contribute to this goal.

  1. The introduction has been corrected to take into account sometimes contradictory suggestions from the reviewers and the editor.

Q: Specify the type of fermentation being discussed and provide more details on the experimental setup, including the incubated mixture and conditions used in the study.

  1. the data suggested by the reviewer were supplemented in the manuscript

Q: Ensure that the reported results are presented with precision. Instead of saying "a decrease in fermentation rates," provide specific numerical values or percentages for better clarity.

R: has been corrected in accordance with the reviewer's suggestions.

Q: Be consistent in the terminology used. For example, "saponite clay" is referred to as "saponit clay" at one point.

R: As suggested, the terminology has been unified throughout the manuscript.

Reviewer 3 Report

Comments and Suggestions for Authors

Dear Authors,

The manuscript titled "The effect of saponite clay on the ruminal fermentation in cows during in vitro studies" is aiming to evaluate the effects of using raw saponite clay as a fermentation modifier to affect the production of enteric methane emissions and volatile fatty acids. To that, an in vitro fermentation study was run by testing three different diets.

The article has several areas of weakness, specially due to the lack of relevant information in Material and Methods, missing controls in Results, lack on concreteness in Introduction and Discussion, inaccuracies in References and editing errors. Major reviewed is needed to justify the relevance of the topic under research in Introduction and Discussion. Also, it is needed a clarifaction of the experimental design applied in Material and Methods by adding the statistical model applied and the factors under study. Moreover, a full description of the donor animals used and details related to the collection of samples are needed to replicate it. Tables need even to be checked to show clearly significant differences between treatments by reviewing the letters. Finally, References need too to be reviewed according to Animals' Guidelines and editing errors have to be also solved before accepting it for publication.

Please, see below a list of specific comments/suggestions to be addressed.

L2 Replace 'in cows' by 'parameters'.

L34 Explain the meaning of the acronym 'FE' before mentioning it.

L52 Replace 'as relative' by 'as the relative'.

L88 Replace 'of prevention' by 'for prevention'.

L89 Replace 'saponite, saponite, and selenite' by 'saponite and saponite plus selenite'.

L91-L92 Replace 'it is recommended that saponite clay as a feed additive/sorbent be fed' by 'saponite clay is recommended as a feed additive/sorbent to be fed'.

L93 Replace 'the body the absence' by 'the body meanwhile provoking the absence'.

L97 Replace 'was' by 'were'.

L101-L105 Provide detailed information about the characteristics of the donor animals used (number, age, parity, body weight, body condition score, total dry matter intake, etc.).

L111-L114 Give more details about the procedure applied for running the experiment (explain how the collection of rumen samples was done).

L121 Replace 'is' by 'was'.

L122 Replace 'is sold' by 'is usually sold'.

L133-L134 Review this sentence and rewrite it.

L155 Explain the meaning of the acronym 'SEM'.

L189 Replace 'At the end of fermentation pH' by 'At the end of the fermentation process, pH'.

L218-L224 Add the statistical model and describe the factors under study.

Table 2: Check letters for showing significant differences among treatments.

L261 Replace 'in respect of' by 'in comparison to'.

Table 3: Check letters for showing significant differences among treatments.

L269-L287 Delete all these sentences. Replace them by other statements more appropriate. Try to focus on the topic of research and be concrete.

References: Check citations according to Animals' guidelines for authors.

Yours sincerely,

Reviewer. 

Author Response

The Authors would like to thank for insightful comments that will improve the quality of our manuscript. We have followed the Reviewer’s suggestions.

Q: The introduction is too long with too much tangentially related information about saponite clay. Please make it more concise.

R: The introduction has been corrected to take into account sometimes contradictory suggestions from the reviewers and the editor.

Q: The article has several areas of weakness, specially due to the lack of relevant information in Material and Methods, missing controls in Results, lack on concreteness in Introduction and Discussion, inaccuracies in References and editing errors. Major reviewed is needed to justify the relevance of the topic under research in Introduction and Discussion.

R: The manuscript has been revised and supplemented with additional information.

Q: Also, it is needed a clarifaction of the experimental design applied in Material and Methods by adding the statistical model applied and the factors under study. Moreover, a full description of the donor animals used and details related to the collection of samples are needed to replicate it.

  1. The Materials and Methods section has been corrected and new information has been added in line with the suggestions of all reviewers

Q:Tables need even to be checked to show clearly significant differences between treatments by reviewing the letters.

  1. The significant differences have been checked and corrected, and appropriate changes have been made to the tables.

Q: Please, see below a list of specific comments/suggestions to be addressed.

L2 Replace 'in cows' by 'parameters'.

L34 Explain the meaning of the acronym 'FE' before mentioning it.

L52 Replace 'as relative' by 'as the relative'.

L88 Replace 'of prevention' by 'for prevention'.

L89 Replace 'saponite, saponite, and selenite' by 'saponite and saponite plus selenite'.

L91-L92 Replace 'it is recommended that saponite clay as a feed additive/sorbent be fed' by 'saponite clay is recommended as a feed additive/sorbent to be fed'.

L93 Replace 'the body the absence' by 'the body meanwhile provoking the absence'.

L97 Replace 'was' by 'were'.

L111-L114 Give more details about the procedure applied for running the experiment (explain how the collection of rumen samples was done).

L121 Replace 'is' by 'was'.

L122 Replace 'is sold' by 'is usually sold'.

L133-L134 Review this sentence and rewrite it.

L155 Explain the meaning of the acronym 'SEM'.

L189 Replace 'At the end of fermentation pH' by 'At the end of the fermentation process, pH'.

L218-L224 Add the statistical model and describe the factors under study.

Table 2: Check letters for showing significant differences among treatments.

L261 Replace 'in respect of' by 'in comparison to'.

Table 3: Check letters for showing significant differences among treatments.

R: All the above-mentioned suggestions have been taken into account

Q: L269-L287 Delete all these sentences. Replace them by other statements more appropriate. Try to focus on the topic of research and be concrete.

  1. The information in this paragraph was written at the request of the editor. However, we have made these paragraphs shorter.

R: All the above-mentioned suggestions have been taken into account

Q: References: Check citations according to Animals' guidelines for authors.

R: References have been corrected in accordance with the journal's requirements

Q: L101-L105 Provide detailed information about the characteristics of the donor animals used (number, age, parity, body weight, body condition score, total dry matter intake, etc.).

R: The study was carried Polish Holstein Friesian cows of the black-and-white variety, approximately 610 ± 26 kg of BW (multiparous cows), at 3.5 ± 0.25  BCS. Dry matter intake (DMI) was 11 kg/d. Fresh feed was delivered twice a day (0600 to 0700 h and 1400 to 1500 h). During the entire trial, the feed offered was adjusted daily to target a refusal rate of 5 to 10%.

Reviewer 4 Report

Comments and Suggestions for Authors

This manuscript descibes an experiment evaluating saponite clay as an additive to alter in vitro fermentation to reduce methane emissions. I have a couple of major concerns.

1. The statistical analysis needs to be better described along with the number of experimental units per treatment so that the reader can understand the analysis.

2. The presentation of gas/methane data is inadequate to accurately interpret what happened in the experiment. My interpretation of the results disagrees with the authors. I think that the saponite clay decreased digestion of organic matter, thus reducing total VFA production, total gas production, and methane production. The gas and methane data need to be presented differently to really determine how the saponite clay affected the fermentation. The saponite clay did affect the proportion of VFA, which may have reduced methane emissions, but likely not to the extent that the current data format shows and the authors seem to have selective bias on which variables with p-value between 0.05 and 0.10 they deem significant and which they do not.  

Specific comments are in the attached PDF.

Comments on the Quality of English Language

English grammar needs moderate revision. Please have English expert review the paper.

Author Response

The Authors would like to thank for insightful comments that will improve the quality of our manuscript. We have followed the Reviewer’s suggestions.

Q: The statistical analysis needs to be better described along with the number of experimental units per treatment so that the reader can understand the analysis.

  1. has been corrected in accordance with the reviewer's suggestions. There are 7 biological reps and each of those was divided into 3 bottles; 7 x 3 = 21 bottles. The 21 bottles were divided among 3 treatments; 21 / 3 = 7 bottles per treatment.

Q: The saponite clay did affect the proportion of VFA, which may have reduced methane emissions, but likely not to the extent that the current data format shows and the authors seem to have selective bias on which variables with p-values between 0.05 and 0.10 they deem significant and which they do not.  

R: The statistical analyses have been rechecked, and the description of the statistical changes presented in tables and manuscript body has been improved.

Q: The presentation of gas/methane data is inadequate to accurately interpret what happened in the experiment. My interpretation of the results disagrees with the authors. I think that the saponite clay decreased digestion of organic matter, thus reducing total VFA production, total gas production, and methane production. The gas and methane data need to be presented differently to really determine how the saponite clay affected the fermentation.

R: In the discussion, there are potential mechanisms accountable for the decrease in methane production subsequent to the utilisation of mineral clay. We concur that one of them might be the mechanism the reviewer suggested.

C: “Rindsig et al. [48] tested nursing cows at risk of acute and subacute ruminal acidosis with 5% and 10% bentonite clay.” This level of clay could alleviate subacute ruminal acidosis simply due to substitution of fermentable carbohydrate

R: No significant changes in rumen pH were noted throughout the trial. The bentonite treatments resulted in a return to the normal rumen acetate-to-propionate ratio. A return to more nearly normal arterial blood acetate levels was also noted with the bentonite treatments. These authors suggest that bentonite may act to slow down the rate of passage through the rumen and thereby affect rumen microbes resulting in increased rumen acetate and decreased rumen propionate production.

Q: information about number of treatments

R: lines 112-118 and 172-177. Briefly, seven biological samples of rumen fluid were collected, and prepared for fermentation experiment. Each biological samples were divided into three groups: Group C (control); Group I and Group II. To determine the fermentation parameters, three technical replicates for each group per one biological sample were carried out using the Ankom system. So, there was 7 biological replicates per treatment group followed by 3 technical replicates for each experimental group (n=21). Totally, in our studies, we analysed 63 samples.

Q: why 30 g? please explain further.

R:  Although not all in vitro systems are perfectly able to mimic the rumen environment, they remain an integral part of reducing the overall cost of animal testing. In vitro testing can be treated as a screening method. Animal research remains indispensable, and in vitro systems will never be able to substitute for the use of animals (Muqier et al., 2023; Wang, 2023; Shaw et al., 2023). To accurately guess how much CH4 will be made in vivo based on in vitro conditions, it is important to think about how complicated the fermentation conditions are in the rumen and how the nutrients move through it (Hatew et al. 2015). In studies that use in vitro fermentation of rumen content, substrates are often used in larger amounts than those that are used directly in cow feed in vivo (Darwin et al., 2022; Sucu et al., 2023; Dhakal et al., 2022). Furthermore, in vitro tests cannot be translated 1:1 into in vivo conditions. Moreover, in vivo, use of clay requires consideration of its chemical makeup when dietary inclusions are made. Our in vitro studies indicate the effect of clay on methane production levels. It was a large dose given at one time.  We hypothesise that, in vivo, a lower level of clay will have a positive effect on the fermentation profile, but with long-term action. The use of high doses in the in vitro studies without side effects in the fermentation profile indicates that the use of clay in the feeding of cows may also be high. Many authors in the in vitro-based analysis that test different feedingstuffs in the conclusions write that the results of the in-vitro incubation should be analysed for long-term effects and effectiveness in vivo (Martínez-Fernández et al. 2013, Vadroňová et al. 2023). Hence, it seems justifiable to employ higher experimental dosages in in vitro investigations.

  1. Wang, M. In Vitro Fermentation. Fermentation 2023, 9, 86. https://doi.org/10.3390/fermentation9020086
  2. Darwin, Tiya Humaira, Ami Muliawati. Identification and characterization of acidosis on in vitro rumen fermentation with feeds based on grass, rice bran, concentrate, and tofu pulp.Journal of Applied Biology & Biotechnology Vol. 10(Suppl 1), pp. 53-58, Mar-Apr, 2022
  3. Dhakal, R.; Copani, G.; Cappellozza, B.I.; Milora, N.; Hansen, H.H. The Effect of Direct-Fed Microbials on In-Vitro Rumen Fermentation of Grass or Maize Silage. Fermentation 2023, 9,347. https://doi.org/10.3390/ fermentation9040347
  4. Sucu, E. In Vitro Studies on Rumen Fermentation and Methanogenesis of Different Microalgae and Their Effects on Acidosis in Dairy Cows. Fermentation 2023, 9, 229. https://doi.org/ 10.3390/fermentation9030229
  5. G. Martínez-Fernández 1, L. Abecia, A.I. Martín-García, E. Ramos-Morales, G. Hervás, E. Molina-Alcaide, D.R. Yáñez-Ruiz In vitro–in vivo study on the effects of plant compounds on rumen fermentation, microbial abundances and methane emissions in goats. 2013, Animal Volume 7, Issue 12, 2013, Pages 1925-1934
  6. Mariana Vadroňová, Adam Šťovíček, Kateřina Jochová, Alena Výborná, Yvona Tyrolová, Denisa Tichá, Petr Homolka & Miroslav Joch. Combined effects of nitrate and medium-chain fatty acids on methane production, rumen fermentation, and rumen bacterial populations in vitro. 2023. Scientific Reports volume 13, Article number: 21961 (2023)
  7. Shaw, C.A.; Park, Y.; Gonzalez, M.; Duong, R.A.; Pandey, P.K.; Brooke, C.G.; Hess, M. A Comparison of Three Artificial Rumen Systems for Rumen Microbiome Modeling. Fermentation 2023, 9, 953. https://doi.org/ 10.3390/fermentation9110953
  8. Hatew, B.; Cone, J.W.; W.F. Pellikaan, S.C. Podesta a, A. Bannink, W.H. Hendriks, J. Dijkstra Relationship between in vitro and in vivo methane production measured simultaneously with different dietary starch sources and starch levels in dairy cattle, Animal Feed Science and Technology Volume 202, April 2015, Pages 20-31
  9. Muqier, X.; Eknæs, M.; Prestløkken, E.; Jensen, R.B.; Eikanger, K.S.; Karlengen, I.J.; Trøan, G.; Vhile, S.G.; Kidane, A. In Vitro Rumen Fermentation Characteristics, Estimated Utilizable Crude Protein and Metabolizable Energy Values of Grass Silages, Concentrate Feeds and Their Mixtures. Animals 2023, 13, 2695. https://doi.org/10.3390/ ani13172695

Round 2

Reviewer 1 Report

Comments and Suggestions for Authors

The author has basically considered the issues mentioned earlier

Author Response

On behalf of all the authors, I would like to thank you for your insightful comments, we believe that thanks to your suggestions we were able to improve the manuscript.

Reviewer 3 Report

Comments and Suggestions for Authors

Dear Authors,

Your manuscript is now suitable for publication at Animals in present form.

Yours sincerely,

Reviewer.

Author Response

(The authors gave the same response as above.)

Reviewer 4 Report

Comments and Suggestions for Authors

The authors have improved the manuscript but there are still some things that need to be addressed. See attached documents.

Comments on the Quality of English Language

The English grammar is improved from the first version, but an English expert should review the paper before acceptance.

Author Response

On behalf of all the authors, I would like to thank you for your insightful comments, we believe that thanks to your suggestions we were able to improve the manuscript.

Q: determined

R: has been corrected in accordance with the reviewer's suggestions.

Q: The author's response to the number of experimental units does not match Line 317. The response indicates 7 bottle per treatment whereas Line 317 indicates n=21 bottle per treatment. Another issue is that 7 biological reps is not divisible by 3 treatments so based on the description given in the response each treatment had only 1 analytic rep per biological rep, thus there are no analytical reps per treatment, only biological reps. This explanation needs to be clarified with Line 317.

Q: This makes much more sense with the number of experimental units. Please include this at Line 228.

R: The paragraph was rewritten as correctly recommended by the reviewer

Lines 214 -224 and 305-306

Seven biological samples of rumen fluid were collected, and prepared for fermentation experiment. Each biological sample was homogenised, and mixed with prewarmed to 39 °C buffer [50] in a 1: 3 ratio (20 ml of rumen fluid to 60 ml of buffer) and transferred to prewarmed glass bottles. Then each biological sample were divided into three groups: Group C (control) received 1 g of a mixture of corn and grass feed (1/1); Group I received 1 g of a mixture of corn and grass feed (1/1) supplemented with 0.15 g of saponite clay; and Group II received 1 g of a mixture of corn and grass feed (1/1) supplemented with 0.25 g of saponite clay. To determine the fermentation parameters, three technical replicates for each group per one biological sample were carried out using the Ankom system. So, there was 7 biological replicates per treatment group followed by 3 technical replicates for each experimental group (n=21). Totally, in our studies, we analysed 63 samples.

The analysis data represent seven biological replicates with three analytical repeats for each biological sample (n=21).

Q: Since neither biological nor analytical rep was included in the model that means individual bottle was the experimental unit. This should be stated plainly.

R: has been corrected in accordance with the reviewer's suggestions.

The statistical model was as follows: Zij = x + fi + eij

where Zij represents the value of each individual observation (individual bottle was the experimental unit), x is the overall mean, fi is the fixed effect of the treatment (different additives; C, I, II), and eij is the residual error.

Q: This appears to be cumulative over time. This should be indicated in the table title or footnote.

R: has been corrected in accordance with the reviewer's suggestions.

“Table 3. The profile of total gas production and cumulative production of CH4 in the investigated treatments. ‘

Information about *cumulative over time measurements was added to table footnote.

Q: This question addresses my concern with using 0.15 and 0.25 g of clay, but does not address the question about using 30 g of microbial cells in Line 292. Please address the original question.

R: Regrettably, the response was omitted unintentionally during the editing process (copy-paste). So, this stoichiometric calculation published by Chalupa was based upon balances applied to rumen fermentation and published by Demeyer and Van Nevel (1975) ; Hungate (1966); and Wolin (1960). Briefly, the ratio of microbial N produced to VFA produced in long simulation experiments  definitely refers to net synthesis and is close to the theoretical 2.1 g N/mol VFA produced.  Additional studies using DAPA as a bacterial marker for bacterial cell growth in the rumen, summarised in a review by Thomas (1973), showed that the cell yield varied from 15.3 to 51.1 g N/kg organic matter fermented. The values are based on different studies using a range of twenty-seven diets. From all of these data, the generally accepted value is 30 g of bacteria formed (growth) per mol of VFA produced during fermentation [Singh et.al]. Therefore, the calculation of Chalupa refers to 30 g of microbial cells per mole of VFA. We have changed the sentence for this part of the text, as shown below.

“The CY parameter was calculated according to a calculation published by Chalupa [56] based on the value of 30 g of bacteria formed per mole of VFA produced and expressed in (g/L)”.

  1. Demeyer, D. L and C. J. Van Nevel. 1975. Methanogenesis, an integrated part of carbohydrate fermentation, and its control. In I. W. McDonald and A. C. I. Warner (Ed.) Digestion and Metabolism in the Ruminant. p. 366. Univ. of New England Publ. Unit. Arrnidale, N.S.W., Australia
  2. Hung, ate, R. E. 1966. The Rumen and its Microbes. Academic Press, NY.
  3. Wolin, M. J, 1960. A theoretical rumen fermentation balance. J. Dairy Sd. 43:1452.
  4. Thomas PC. Microbial protein synthesis. Proc Nutr Soc. 1973 Sep;32(2):85-91.
  5. Singh UB, Verma DN, Varma A, Ranjhan SK. The relationship between rumen bacterial growth, intake of dry matter, digestible organic matter and volatile fatty acid production in buffalo (Bos bubalis) calves. Br J Nutr. 1977 Nov;38(3):335-40. doi: 10.1079/bjn19770098.